# Structural activity prediction models recover known kinase binding modes

**Michael Backenköhler** [* 1]   **Joschka Groß** [* 2]   **Paula Linh Kramer** [1]   **Verena Wolf** [2]   **Andrea Volkamer** [1]

**Structural ML for kinase drug discovery**   Drug discovery pipelines nowadays rely on machine learning models to explore and evaluate large chemical spaces. Especially in context of small molecular ligands for protein targets, three-dimensional structural models are a natural representation. While such information is beneficial, not enough protein-ligand (PL) complex structures are available to train models on these expressive representations. Addressing kinase protein targets specifically, this issue can be tackled by generating *in silico* kinase-ligand complex data using template docking for the kinase compound subset of available ChEMBL assay data (Schaller et al., 2023). The docking is based on a suitable "template" complex with a structurally close ligand bound to the same target. We leveraged the strong performance of template docking on this protein class to create kinodata-3D – a large *in silico* structural dataset of $\sim 120,000$ PL complexes. This data was shown to indeed aid machine learning models leading to a statistically significant improvement of binding affinity prediction compared to baselines without access to the 3D structure. Details on the dataset creation and the comparative affinity prediction study are given in previous work (Backenköhler et al., 2024).

**Explaining binding affinity predictions**   Our work builds on these structural binding affinity models trained this large dataset of *in silico* complexes. The binding affinity prediction model is an E(3)-invariant GNN that takes a complex graph as an input. In this work, we are interested in understanding what aspects of protein-ligand interaction the model learned from the docking data. Such insight can be gained by observing how model outputs change based on targeted perturbations to its input (Ivanovs et al., 2021).

The complex graph is composed of atoms with two types of edges: covalent bonds and spatial edges that are added due to closeness of atom pairs. We focus our attention on how different parts of the protein structure influence binding affinity prediction. To this end, we mask single residues from the protein-ligand graph and compare the masked prediction against the reference on the entire protein. The method is summarized in Figure 1.

As protein-ligand interactions are typically analyzed on the level of residues, we cut all spatial PL edges for each residue of the 85 binding pocket residues separately. This way we observe the effect of message passing between a single residue and the ligand component. We observe the largest changes in prediction for residues commonly associated with ligand binding such as the hinge and DFG region of the binding pocket (Kanev et al., 2020) (see Figure 2). We further observe that outside of these regions removal of spatial edges has a low impact on prediction. This indicates that the model successfully learned common binding mechanisms.

**Prospective evaluation**   Having seen that simple methods can be used to recover import patterns underlying the kinase binding mechanisms we hope to enhance our XAI approaches and the analysis of their respective explanations for the sake of guiding the exploration of potentially novel kinase binding mechanisms. It allows users to understand the processes that influence model predictions, enabling error detection and a more detailed understanding of a model's ability to make reasonable predictions. This transparency may also support scientific discovery by elucidating complex patterns and relationships within data.

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

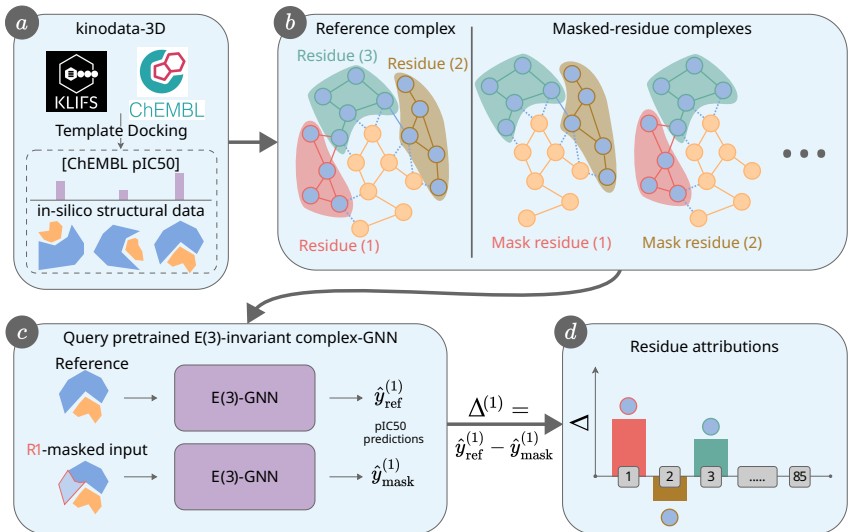

*Figure 1.* Residue masking: (a) PL complexes are generated using template docking. An E(3)-invariant GNN is trained for binding affinity prediction on the resulting dataset. (b) We occlude each pocket residue once by masking the complex graph. (c) The prediction on the unmasked reference is compared to the masked prediction. (d) The resulting changes are analyzed with respect to their biochemical interpretation and known binding mechanisms in kinases.

Salentin, S., Schreiber, S., Haupt, V. J., Adasme, M. F., and Schroeder, M. Plip: fully automated protein–ligand interaction profiler. *Nucleic acids research*, 43(W1):W443–W447, 2015.

Schaller, D., Christ, C. D., Chodera, J. D., and Volkamer, A. Benchmarking cross-docking strategies for structure-informed machine learning in kinase drug discovery. *bioRxiv*, 2023. doi: 10.1101/2023.09.11.557138. URL https://www.biorxiv.org/content/early/2023/09/14/2023.09.11.557138.

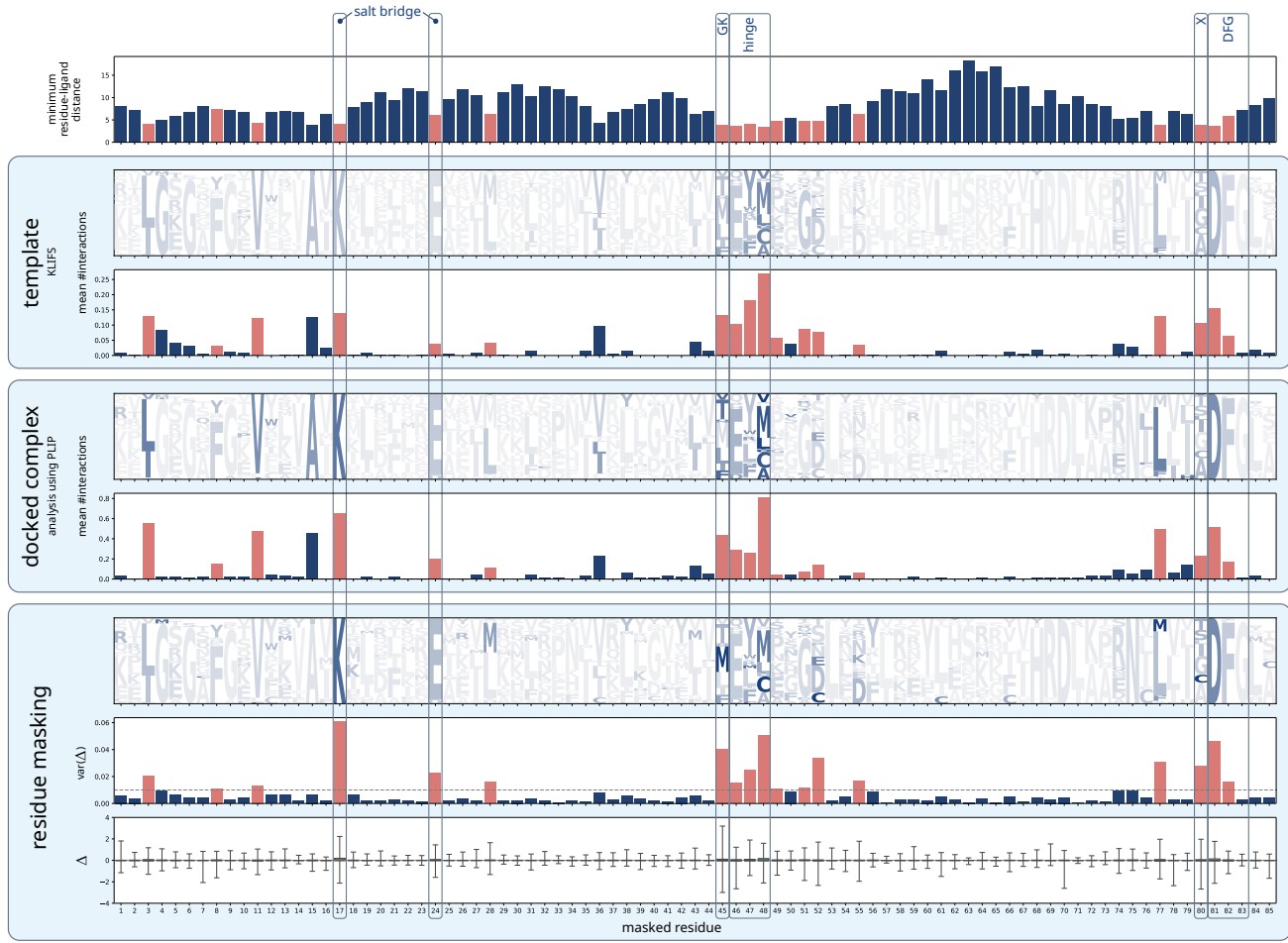

*Figure 2.* Comparison of residue importance. The upper plot shows average minimum distance between the residue on a given position of the pocket sequence to the ligand molecule. The *template* row summarizes average importance of residues for binding. The importance is given in terms of the mean number of PL interactions in the template complex from KLIFS (Kanev et al., 2020). Additionally, prevalence of residue types at different positions is indicated by residue letter size. The darker the letter, the more interactions are present on average. The *docked complex* row shows similar data for the docked complex aggregating interaction analyses with PLIP (Salentin et al., 2015). The bottom section summarizes results of the *residue masking* procedure (this work). The influence are given in terms of the variance of the difference between the reference and the masked prediction $\Delta$.