# OpenReview forum: "Structural activity prediction models recover known binding modes (Poster abstract)"
_ICML.cc/2024/Workshop/ML4LMS — ML4LMS Poster_

### Official Review · ML4LMS · 2024-06-06

**Rating:** 7
**Confidence:** 4

**Review:**

The authors present a poster abstract on early work on using an E(3)-invariant GNN, introducing covalent bonds and spatial edges (that convey information on the closeness of atom pairs). They then probe the influence of these edges by removing them from the input and observing the change in the predicted binding affinity. I have one minor comment that the authors could address in the presented poster directly: "Prospective evaluation" is typically held in chemical space away from training data. Can the authors show that this method can extrapolate to different protein targets, and/or scaffolds in chemical space where the original model was not trained on?

---

### Official Review · ML4LMS · 2024-06-11
**Interesting ideas, but limited results to evaluate**

**Rating:** 6
**Confidence:** 3

**Review:**

The work proposes using synthetic kinase-ligand complex data (obtained via docking) to train a binding affinity prediction model. After training, the authors probe the learned mechanisms of the model by removing spatial edges between the ligand and certain residues of the binding pocket, observing that model performance is most changed when edges connecting ligand atoms to residues that are known to be important (e.g. in the hinge region) are removed.

I find this work potentially very interesting. However, the submission is only an abstract, and there are essentially no concrete results to review. Despite this, I would tentatively recommend acceptance, because I believe the topic could yield useful discussion at the workshop. However, I encourage the authors to include additional evaluations. For example, I am very curious to see how the model behaves on inactive ligands, given notoriously high false-positive rates associated with docking, and the fact that the crystal templates used to dock presumably contain mostly active ligands.

---

### Official Review · ML4LMS · 2024-06-11
**Review of Structural activity prediction models recover known binding modes (Poster abstract)**

**Rating:** 5
**Confidence:** 5

**Review:**

In this poster abstract, the authors describe a new in silico dataset called Kinodata-3D. Kinodata-3D appears to be a large dataset comprised of known kinase ligands from ChEMBL, docked using template docking. The authors claim that using this data, they were able to train a GNN to predict protein-ligand binding and to probe the influence of specific residues on the receptor to potentially explain predicted affinity. They also claim that their model can recapitulate key motifs involved in protein-ligand binding, such as the DFG and hinge.

While the dataset and the results of their trained GNN are interesting, it is challenging to properly evaluate this abstract due to the lack of figures or results to verify the abstract's claims. Additionally, the following information would be beneficial:

How large is the Kinodata-3D dataset?
What QC and filtering steps were taken to ensure that the data produced via template docking is of high quality?
How different are the effect sizes of the hinge and DFG motifs on binding affinity prediction in a model trained on just crystal structures versus Kinodata-3D?
More information on the training/test/validation splits of the GNN model, and general model performance on predicting ligand affinity in terms of typical metrics (Spearman correlation, RMSD, etc.).